# An Overview on Immunogenic Cell Death in Cancer Biology and Therapy

**DOI:** 10.3390/pharmaceutics14081564

**Published:** 2022-07-27

**Authors:** Mosar Corrêa Rodrigues, José Athayde Vasconcelos Morais, Rayane Ganassin, Giulia Rosa Tavares Oliveira, Fabiana Chagas Costa, Amanda Alencar Cabral Morais, Ariane Pandolfo Silveira, Victor Carlos Mello Silva, João Paulo Figueiró Longo, Luis Alexandre Muehlmann

**Affiliations:** 1Faculty of Ceilandia, University of Brasilia, Brasilia 72220-275, Brazil; mosarcr@gmail.com (M.C.R.); joseavmorais@gmail.com (J.A.V.M.); rayaneganassin@hotmail.com (R.G.); giulia_rosa12@hotmail.com (G.R.T.O.); fchagas16@gmail.com (F.C.C.); 2Laboratory of Nanobiotechnology, Department of Genetics and Morphology, Institute of Biological Sciences, University of Brasilia, Brasilia 70910-900, Brazil; amandaalencarcabral@gmail.com (A.A.C.M.); pandolfo.ariane@gmail.com (A.P.S.); victor@spctm.com (V.C.M.S.); jplongo82@gmail.com (J.P.F.L.)

**Keywords:** DAMPs, immune system, chemotherapy, photodynamic therapy, radiotherapy, immunotherapy

## Abstract

Immunogenic cell death (ICD) is a modality of regulated cell death that is sufficient to promote an adaptive immune response against antigens of the dying cell in an immunocompetent host. An important characteristic of ICD is the release and exposure of damage-associated molecular patterns, which are potent endogenous immune adjuvants. As the induction of ICD can be achieved with conventional cytotoxic agents, it represents a potential approach for the immunotherapy of cancer. Here, different aspects of ICD in cancer biology and treatment are reviewed.

## 1. Introduction

Both the innate and the adaptive branches of the immune system are involved in the elimination of malignant cells. Generally, for the successful elimination of tumors by immunity, the in situ presence of immunoadjuvants and antigens, as well as a non-immunosuppressor tumor microenvironment, are essential.

Tumor cells commonly express neoantigens, which are different from any other normal protein in the host, or tumor-associated antigens, which are expressed in an unusual tissue or in aberrantly high amounts [1,2]. As a result, one important feature of cancers is their ability to escape immunity, as the immune system is crucially involved in the defense against the development and progression of malignant cells [3]. 

In certain situations, the immunogenicity of tumors can be enhanced by increasing the local release and exposure of endogenous immunoadjuvants, such as damage-associated molecular patterns (DAMPs). The release of these molecules with a well-defined, immune-activating pattern is observed in a regulated cell death (RCD) modality known as immunogenic cell death (ICD). In this review we discuss how ICD can help to trigger or boost immune responses against cancer.

## 2. A Historical Overview on Cancer Immunotherapy

Activating or boosting adaptive immune responses against tumors has become a main pillar in the treatment of different cancers. Currently, immune-checkpoint inhibitors, chimeric antigen receptor (CAR) T cells, dendritic cell (DC)-based vaccines and immunostimulatory cytokines have been successfully used in clinical practice [4]. Different immune-checkpoint inhibitors approved by the FDA have significantly improved the treatment of advanced-stage melanoma, non-small-cell lung carcinoma, kidney carcinoma, urothelial carcinoma, hepatocellular carcinoma, and others [4,5]. 

The first evidence that the host immune system could be therapeutically targeted to treat cancer was brought to light in the 19th century, when erysipelas was reported to have an influence on tumor progression. This erythematous infection of the skin was relatively common in post-surgery patients given the poor sanitary conditions of surgery at that time. In 1867, the German physician Wilhelm Busch reported that a malignant tumor disappeared after the patient had contracted this infection [6]. 

In 1883, the German surgeon Friedrich Fehleisen demonstrated that the etiologic agent of erysipelas was *Streptococcus pyogenes* [7]. In his experiments, Fehleisen produced erysipelas by inoculating patients with pure cultures of *S. pyogenes*. On one occasion, out of six patients with inoperable malignancies who reacted to this inoculum, three showed only a slight change in their tumors. The other three patients, however, exhibited significant changes in the evolution of the disease. The fibrosarcoma nodules of one patient thus treated were reported to have disappeared, while two other patients with breast carcinoma had tumor remission. Similar clinical experiments were performed by other surgeons at that time [8].

In 1891, the American surgeon William Bradley Coley published an article citing the work of Fehleisen and his own observations on the curative effect of erysipelas upon malignancies [9]. Coley injected patients with *S. pyogenes*, inducing erysipelas on purpose. Some of these patients, after having recovered from local inflammation and fever, exhibited reduced tumor size or were even cured of the malignancy. According to Coley, some promptly responded to this therapy upon the first injection, while others needed several doses before some clinical improvement could be observed. As there were cases of death after the injection, Coley decided to inactivate the bacteria with heat before injecting them, in a preparation which was named Coley’s toxin. The use of this bacterial preparation gave good results in bone and soft-tissue sarcomas [10]. This was one of the first anticancer immunotherapies described in the scientific literature.

Despite the reported success, many were quite skeptical about the results described by Coley. As recognized later, his work was probably the first formal study on immunotherapy and presented a rationale for inducing immune responses against tumors, but the lack of knowledge on immune mechanisms at the time his first patients were treated was a huge barrier to the acceptance of this approach [10,11]. However, as immunology advanced, many oncologists began to support the use of Coley’s toxin. In 1935, the prestigious surgeon Ernest Amory Codman asserted that the results described by Coley were solid scientific evidence [10,12]. Codman was particularly impressed with six cases registered by Coley of five-year cures of patients with Ewing’s sarcoma injected with the toxin [12]. Codman suggested that the increased production of lymphocytes induced by Coley’s toxin might explain the “occasional miracle which follows this treatment”.

In 1909, the renowned German scientist and physician Paul Ehrlich suggested that the immune system played an important role in eliminating cancer cells, speculating that it might be involved in suppressing the development and progression of carcinomas [13]. Later, as the role of the immune system in distinguishing self from non-self became clear, Frank Macfarlane Burnet developed the concept of immunosurveillance [14,15].

As Burnet wrote [15], cancer cells grow free from the normal control exercised by the organism as a whole. Cancer is then the consequence of the breakdown in one or more aspects in this control mechanism that holds the multitude of cells of an organism working together as a single functioning unit [15]. One key part of this normal control was the immune system, which is normally able to identify and to mount an effective reaction against tumor-specific or -associated antigens, eventually eliminating most of the potential cancer cells before they become a clinically apparent tumor [15].

In a favorable immune context, tumor antigens are processed and presented by antigen presenting cells (APCs), triggering the activation of CD4^+^ and CD8^+^ T cells, ultimately eliminating the tumor. However, malignant cells can suppress their own immunogenicity, thus avoiding being detected by the immune system. Cancer cells can, for instance, upregulate the expression of programmed cell death ligand 1 (PD-L1) molecules, present defects in the antigen presentation machinery, recruit immunosuppressive cells, such as myeloid-derived suppressor cells (MDSC) and T regulatory (Treg) cells, and lead to direct or indirect secretion of TGF-β and IL-10 [5,16].

The development of an immunoevasive phenotype is the result of a long, complex and dynamic interaction of transformed host cells and the immune system in what is known as cancer immunoediting [16,17]. This process can be divided into three stages: elimination, equilibrium and escape. If elimination is successful, the immune system eradicates aberrant cells and prevents the continuation of carcinogenesis. This event represents the fulfillment of immunosurveillance and is highly dependent on the immunogenicity of the abnormal cells. If the elimination is not complete, however, an equilibrium stage may occur, i.e., although these cells persist, they are not able to freely proliferate, and do not generate clinically apparent tumors. Occasionally, some of these aberrant cells may develop more efficient immune evasion abilities, then becoming poorly immunogenic or non-immunogenic. These cells eventually escape the immune system and generate a growing tumor mass [17,18].

Therefore, changing the balance towards the host immune system could be used therapeutically against cancers. In the 20th century, many advances in immunology contributed to the development of modern immunotherapy [19]. Some of these advances are particularly noteworthy: (i) the identification and demonstration of the role of T lymphocytes in animal models [20]; (ii) the demonstration of the presence of dendritic cells in peripheral lymphoid organs [21]; and (iii) the identification of natural killer cells (NK cells) [22]. Many approaches to target the immune system as an anticancer strategy followed, such as the use of cytokines, vaccines, adoptive cell therapies and antibodies against immune checkpoints [11].

The most recent milestone in modern immunotherapy, the discovery of immune checkpoint inhibitors, led to the development and approval by the FDA of anti-PD-1/PD-L1 and anti-CTLA-4 antibodies, which proved to be effective against melanoma and other different tumors [23].

Currently, many studies show that it is possible to increase the immunogenicity of tumors by triggering specific cell death modalities in cancer cells. In that regard, it has been observed that specific cytotoxic treatments, such as anthracycline-based chemotherapy [24] radiotherapy [25] and photodynamic therapy (PDT) [26,27], can induce immunogenic cell death (ICD), which can render cancers more efficient at triggering or boosting tumor antigen-specific immune responses. As this event can lead to the elimination of the primary tumor, and of occasional antigen-sharing metastatic foci as well, the induction of ICD has been suggested as a potential immunotherapeutic approach.

## 3. Immunogenic Cell Death

As defined by the Nomenclature Committee on Cell Death, ICD is a modality of RCD [28,29]. This implies that ICD activation depends on signaling transduction programs, and can thus be triggered or modulated by drugs and genetic components [28].

ICD differs from other RCD types, such as necroptosis, ferroptosis and pyroptosis, not only by the conditions under which it is triggered, but also by the fact that it activates an adaptive immune response in immunocompetent syngeneic hosts against antigens expressed by the dying cell [28]. The cells undergoing ICD exhibit morphological and molecular hallmarks of apoptosis with a well-defined pattern of release and exposure of DAMPs [29]. Thus, cells at ICD exhibit the two conditions necessary for eliciting an adaptive immune response: antigenicity and adjuvanticity [28,29].

DAMPs form a group of different molecules that normally perform structural and metabolic functions in living cells unrelated to their immune functions during ICD [30]. However, when emitted with a specific temporospatial profile during ICD, DAMPs are able to trigger or boost antigen-specific immune responses [31,32]. The activation of pattern recognition receptors (PRR) by DAMPs results in the maturation of dendritic cells (DCs) and the activation of CD4^+^ and CD8^+^ T cells [33], as represented in Figure 1. Different aspects of the activation of adaptive immune responses are fulfilled by the different DAMPs involved in ICD.

The time profile of the release and exposure, as well as the specific actions performed by the DAMPs, orchestrates the attraction, phagocytic activity and maturation of APCs in the tumor bed. The simple presence of a single DAMP or just a couple of them in the vicinity of tumor cells is generally not enough for the initiation or boosting of a cytotoxic, effective anticancer immune response. The absence of calreticulin or ANXA1, for example, is known to severely limit immune responses against tumor cells [34,35,36]. Moreover, the tumor microenvironment must permit immune cells to be activated and to perform their roles properly in order for DAMPs to exert their immunoadjuvant effects.

Specific conditions are known to induce ICD, such as chemotherapy [37,38,39], radiotherapy [40,41] and PDT [26,27,42,43,44,45]. Parameters such as the protocol of application and the drug used in these treatments are crucial to determine whether ICD is induced or not. In the case of chemotherapy, for instance, ICD can be induced by drugs such as mitoxantrone, oxaliplatin and cyclophosphamide, but not by cisplatin, etoposide and mitomycin C. In the case of PDT, the type and concentration of the photosensitizer, as well as the irradiation regimen, are key factors [26,27]. 

## 4. Endoplasmic Reticulum Stress

Different ICD-inducers act as stressors of the endoplasmic reticulum (ER) of target cells [31,33,46]. Indeed, the ER has been linked by many studies to programmed cell death [46,47,48]. Stressing conditions, such as oxidative stress and hyperthermia, can impair the folding of proteins. Unfolded proteins bind the luminal ER chaperone GRP78/BiP (Figure 2), triggering the activation of three ER transmembrane proteins: inositol-requiring enzyme 1α (IRE1α), protein kinase RNA-like endoplasmic reticulum kinase (PERK), and activating transcription factor 6 (ATF6). The downstream cellular events following the activation of these ER sensors can either restore normal ER metabolism or induce cell death [48].

The activated IRE1α dimerizes and is autophosphorylated, becoming able to catalyze the unconventional, cytosolic splicing of the mRNA for the transcription factor X box-binding protein 1 (uXBP1) to sXBP1, which is then translated. The XBP1 is a key protein in UPR, activating the transcription of different genes that can reinstate normal ER metabolism [49]. The PERK protein also dimerizes and auto-phosphorylates, activating its kinase domain to phosphorylate the eukaryotic initiation factor 2α (eIF2α) [50,51]. The phosphorylation of eIF2α is the key event in the integrated stress response (ISR), which is a part of the ER stress response, and is common to ICD induced by different cytotoxicants [34]. Thus, the presence of phosphorylated eIF2α is a hallmark of ICD and can be used as a biomarker of ISR in cell cultures and in biological samples [52].

Phosphorylated elF2α triggers the selective translation of the activating transcription factor (AFT4), which activates the expression of genes involved in protein folding, amino acid metabolism and regulation of oxidative stress [53]. Also, ATF6 is translocated to the Golgi apparatus, where it is then cleaved to ATF6N, releasing its transcription factor moiety that activates the transcription of genes for chaperones, XBP1, C/enhancer binding protein-homologous protein (CHOP), and others [48,54].

The UPR is important for cells under stress to survive and restore homeostasis. The activation of this response is often observed in cancer cells, as the tumor is often a stressing environment, with acidic pH and hypoxia, for instance. Thus, the phosphorylation of eIF2α is frequently observed in cancer cells and plays an important role in tumor growth, invasion, and angiogenesis [55]. In this situation, tumors can cope with the constant stress and progress.

However, if the UPR is unable to restore the ER protein folding capacity, then cell death is triggered. In this context, the pro-apoptotic factor CHOP plays a key role [55,56,57]. The PERK/eIF2α/ATF4 pathway activates the production of CHOP [53], which activates the expression of the BH3-only protein Bim. This leads to the activation of Bax/Bak and the release of cytochrome C to the cytosol, with the consequent induction of apoptosis [56,58]. It is worth mentioning that CHOP also acts on the Bcl2 protein (autophagy activation factor), inhibiting its action [59,60]. IRE1α and ATF6 will also activate CHOP via fragments of XBP1S or ATF6N, thereby triggering apoptosis [60,61].

An intense stress in the ER not only activates apoptosis but also promotes the exposure of ER chaperones, such as calreticulin and HSP70, and other DAMPs involved in ICD [62]. The main DAMPs are discussed in the next section. 

## 5. Damage-Associated Molecular Patterns

Processes associated with ICD result in the emission of DAMPs, such as HSP70, HSP90, calreticulin, HMGB1, ATP, type I IFN, cancer cell-derived nucleic acids, ANXA1, and others [29,31,63]. The recognition of DAMPs by cells of the immune system triggers intracellular signaling pathways involved in responses to injury [33]. Frequently, many of these DAMPs perform specific functions in cells not related to immunity. Nevertheless, outside the cells, they can activate various immune cell receptors from the family of PRR. They are also capable of potentiating pro-inflammatory effects such as the maturation and activation of antigen-presenting cells, such as dendritic cells and macrophages, which ultimately activate T cells (CD8^+^), creating an anti-tumor immunity [32].

The DAMPs are emitted by different mechanisms, such as the active exposure on plasma membrane (e.g., calreticulin, HSP70 and HSP90), released as end-stage degradation products (e.g., DNA and RNA), and secreted to the extracellular medium (e.g., ATP, HMGB1, uric acid, IL-1α and other pro-inflammatory cytokines) [33,64].

A well-studied DAMP is calreticulin, a soluble protein found mainly in the ER. Its main functions in this organelle are the buffering of Ca^2+^ [65], contributing to the homeostasis of this cation, and the lectin-like chaperone activity, essential to the folding of N-glycosylated proteins [66]. During ICD, it is exposed on the plasma membrane even before phosphatidylserine [67] and acts as an eat-me signal, inducing antigen-presenting cells to phagocytose the target cell [68]. Although calreticulin is essential to the immunogenicity of cells undergoing ICD, its presence alone on the cell surface is not enough for starting an immune response [36]. The presence of other DAMPs, therefore, is crucial for the immunogenicity of ICD. Other chaperones exposed on the plasma membrane in ICD are HSPs, which also act as eat-me signals for phagocytes [69,70]. 

In cancer cells undergoing ICD, type I interferons (IFN-Is) are released by mechanisms involving the detection of endogenous nucleic acids [28], i.e., dsRNA by TLR3 [71], or dsDNA by cyclic GMP-AMP synthase (cGAS) [72]. The release of this DAMP has been shown to be crucial for the success of the ICD-inducing anthracycline-based chemotherapy [71]. IFN-Is are potent immunostimulatory proteins with significant anticancer effects, and they have been used to treat chronic myeloid leukemia, acute lymphoblastic leukemia, multiple myeloma, melanoma, Kaposi’s sarcoma, and renal cell and bladder carcinoma [73]. It is notable that the nucleic acids released by cancer cells at ICD are also taken up by DCs and other immune cells, triggering potent IFN-I responses [28].

ANXA1 is also among the crucial DAMPs in ICD. This protein is expressed in granules of different cells, such as neutrophils, eosinophils and monocytes, and is found in different organs and tissues, such as the lungs the bone marrow and intestine [74,75]. Like other members of the annexins superfamily, ANXA1 binds phospholipids and can thus affect eicosanoid production [74]. It has an important role in the regulation and resolution of inflammation. Although ANXA1 has been described as a negative regulator of innate immunity, with neutrophils being its main target [74], it is known to be passively released by cells undergoing ICD, then activating the migration of APCs towards the dying cells, facilitating their engulfment and processing [34]. The importance of this DAMP in tumor immunology can be noted by the fact that the low expression of ANXA1 is often associated with poor DCs and T lymphocyte infiltration in the tumor bed and a higher ability of different human cancers to escape immunity [35].

Another DAMP released to the extracellular medium is ATP [43,76]. Extracellular ATP operates as a strong chemoattractant and promotes not only the recruitment of immune cells but also their maturation [77,78]. In addition to serving as one of the main sources of intracellular energy, ATP also acts in the extracellular signaling mechanisms [67]. ATP is secreted in response to the cytotoxic and cytostatic effects of some aggressive agents such as chemotherapy drugs [46]. ATP release occurs during the process of apoptosis, and different mechanisms can lead to the release of ATP from the intracellular to the extracellular environment, such as the caspase-dependent activation of Rho-associated, coiled-coil containing protein kinase 1 (ROCK1)-mediated, myosin II-dependent cellular blebbing, as well as the opening of pannexin 1 (PANX1) channels, which is also triggered by caspases and autophagy [79,80,81]. Of these three, opening pannexin 1 channels and autophagy appear to occur in ICD.

During the formation of apoptotic bodies, caspases 3 and/or 7 act in the cleavage in the C-terminal portion leading to the opening and activation of PANX 1, then releasing the ATP into the extracellular region [82]. In autophagy, the cytoplasmic constituents are degraded in double-membrane organelles, and the autophagosomes are subsequently fused with lysosomes, resulting in the degradation of the autophagocytic content by acid hydrolases and recycling towards energy metabolism or anabolic reactions, releasing the ATP for the extracellular portion [83]. ATP binds the purinergic receptors P2RY2 and P2RX7 of DCs, resulting in the influx of K^+^ and Ca^2+^ ions, activating the inflammasome NLRP3, followed by activation of caspase-1, consequently stimulating the proteolytic maturation and secretion of interleukin 1 (IL-1) and interleukin 18 (IL-18), contributing to the immunogenic response [84]. 

HMGB1 is a non-histone nuclear protein that, once secreted, acts as an essential DAMP in the activation of DCs by ICD [76,85,86,87]. Its release occurs sometime after apoptosis [63,86]. It can also be released by cells of the innate immune system without the need for cell death in response to pathogens or secreted by cells in response to some damage in the late phase of apoptosis—necrosis [86,88,89]. HMGB1 activates toll-like receptors (TLR)4 and 2, and stimulates the production of pro-inflammatory factors by DCs [90], strongly contributing to the immunogenicity of ICD [33,91]. 

The exposure of calreticulin and the release of ANXA1, ATP and HMGB1 result in the attraction and maturation of DCs in the tumor microenvironment [91,92]. Once there, the DCs phagocytose the tumor cells and release pro-inflammatory mediators in the tumor, such as IL-1β, IL-18, IFN-γ, among others. Activated DCs also present the tumor antigens, culminating mainly in the activation of CD8^+^ cytotoxic T lymphocytes, which are fundamental in the activation of the tumor retraction immune antitumor response [33,43]. The effects of the DAMPs discussed in this section are summarized in Table 1.

## 6. ICD and DAMPs in Cancer Therapy

The literature on ICD in cancer biology has been significantly expanded over the last years, providing new possibilities for cancer treatment. Firstly, the discovery that ICD can be induced by certain classical anticancer agents has made it clear that some treatment regimens should be designed not only to directly eliminate tumor cells, but also to optimize their capacity to induce immune responses against tumor antigens. Secondly, ICD inducers can be used in combination with immunotherapeutics, such as immune checkpoint inhibitors. 

Recently, Ganassin et al. (2022) demonstrated that curcumin, a polyphenol obtained from turmeric, causes ER stress and ICD in colorectal adenocarcinoma CT26 cells. The authors observed that, when treated with curcumin, these cells exhibited an initial increase of intracellular Ca^2+^, which was followed by the activation of XBP1, a protein involved in the UPR response. The ER stress initiated by curcumin was accompanied by the induction of ICD.

Regarding chemotherapeutics, different drugs have been found to induce ICD in preclinical studies [36]. Moreover, chemotherapeutics such as doxorubicin, mitoxantrone, idarubicin, cyclophosphamide, bortezomib and oxaliplatin have already been clinically tested as ICD inducers, mainly in combination with immunotherapeutics [39]. For instance, the combination of doxorubicin, an ICD inducer, with the PD-1-targeting immune-checkpoint blocker, nivolumab, resulted in 35% objective response rates (ORRs) in the treatment of metastatic triple negative breast cancer patients [39,97]. When nivolumab was combined with cisplatin, a non-ICD inducer chemotherapeutic, ORRs were lower (23%) [97]. A number of other clinical studies are already under way to test this immunostimulatory combination using not only immune-checkpoint blockers, but also CAR-T cells, DC-based vaccines, immunostimulatory cytokines, and others [39].

Interesting results of the Phase II study published by Bota et al. (2018) also show that ICD-inducers can improve clinical outcomes in glioblastoma (WHO grade IV astrocytic glioma) immunotherapy [98]. The authors used an allogeneic/autologous therapeutic glioblastoma vaccine (ERC1671, Gliovac), which is a mixture of inactivated tumor cells and lysates of tumor cells derived from the treated patient and three other glioblastoma patients, combined with recombinant colony stimulating factor 2 (CSF2, best known as GM-CSF, used to activate immune responses). The patients received a short regimen of low-dose cyclophosphamide (50 mg/day for 4 days) prior to vaccination. Cyclophosphamide is an ICD-inducer, used in this case, according to the authors, for relaxing the immunosuppressive environment. Indeed, low-dose cyclophosphamide has been reported to suppress Treg cell response, helping to promote DC expansion and antitumor cytotoxic T cell-mediated response [99]. The protocol of treatment described by Bota et al. (2018) also included the treatment of these glioblastoma patients with bevacizumab, a monoclonal antibody specific to VEGF. The results showed increased survival for the patients thus treated (12 months vs. 7.5 months for patients receiving bevacizumab only). Although the authors did not discuss the occasional contribution of ICD to the outcomes, the benefit observed with this protocol makes it worth investigating the possible contribution of ICD in preclinical and clinical models, which could help to improve this combinatory therapy in the future.

Radiotherapy has been shown to induce ICD as well. In clinical practice, radiotherapy and chemoradiotherapy were shown to induce the exposure of the ICD-related DAMPs calreticulin, HSP70 and HMGB1 when applied either as a pre- or post-operative protocol of different cancers [100]. The results, however, are still contradictory. A possible cause for this can be the different experimental settings used and insufficient data regarding ICD hallmarks.

Lämmer et al. (2019) investigated the expression of cytosolic HSP70 in tumor tissue of 60 patients diagnosed with primary glioblastoma. As discussed before in this review, HSP70 is released during ICD, acting as a DAMP. In this study [101], the tumors were surgically resected and patients were then treated with radiotherapy and temozolomide chemotherapy. The progression-free survival and overall survival were significantly longer in patients exhibiting a higher expression of cytosolic HSP70. The authors hypothesized that this result may be due to the induction of ICD by the combination of radiotherapy and chemotherapy given to patients.

Rothammer et al. (2019) analyzed the concentration of HSP70 in the serum of 40 breast cancer patients, who received breast-conserving surgery and adjuvant radiotherapy [102]. The authors observed that patients with higher serum concentrations of HSP70 had an increased probability of developing contralateral recurrence or metastases within two years of receiving radiotherapy.

Protocols based on the combination of radiotherapy and immunotherapeutics have also been tested, and it remains unclear if patients benefit from radiotherapy-induced ICD, as the clinical data are contradictory. For instance, in the treatment of rectal cancer, the increase in HMGB1 was associated both with poorer [103] and with better [104] responses to chemoradiotherapy.

Hongo et al. (2015) studied 75 patients with lower rectal cancer who were treated with preoperative chemoradiotherapy, consisting of radiotherapy (1.8 Gy × 28 fractions) and chemotherapy with a 5-fluorouracil (FU) prodrug (300 mg/m^2^/day) and leucovorin (75 mg/day). After chemoradiotherapy, tumors were surgically resected and analyzed for their expression of HMGB1. The patients with a higher expression of HMGB1 had a poorer response to chemoradiotherapy [103]. In the study by Huang et al. (2018), the presence of HMGB-1 in the cytosol, resulting from its translocation from the nucleus, was analyzed in the samples of locally advanced rectal cancer from 89 patients. The authors reported that patients whose cancer exhibited cytosolic HMGB-1 before neoadjuvant chemoradiotherapy had a better clinical outcome.

PDT has also been shown to induce ICD. This therapy is based on the pre-treatment of target cells with a photosensitizer, which is next photoactivated to generate reactive species and cause oxidative stress in situ [105]. As a consequence of the photoreactions thus induced, reactive species are produced, which can overwhelm the antioxidant defenses of the cell, generating oxidative stress and cell death [26]. This approach has been shown to induce ICD in specific conditions. The type and concentration of the photosensitizer [27,106], as well as the light dose [26], can significantly affect its ICD-inducing capacity. Moreover, oxygen supply in the target tissue can also affect PDT outcomes.

Doix et al. (2019) reported that the combination of a therapeutical vaccine based on DCs stimulated with PDT-killed cells and radiotherapy can delay the development of experimental squamous cell carcinoma in mice. Firstly, the authors showed that the non-porphyrinic photosensitizer OR141, at low doses, induced ICD in SCC7 squamous cell carcinoma cells in vitro. These PDT-killed SCC7 cells were then used to prime and stimulate the maturation of DCs in vitro, which were then employed as a therapeutic vaccine against xenografts of SCC7 cells developed in the flank of C3H mice. This DC vaccine was subcutaneously injected three times at one-week intervals near the tumor-draining lymph node. One week later, radiotherapy mediated by a Cesium-137 γ-ray irradiator was applied onto the tumors. Occasionally, a fourth injection of the DC vaccine was performed at the time of radiotherapy application (peri-radiotherapy). It was observed that the tumor growth was delayed only when the DC vaccine was applied during the peri-radiotherapy period.

Rodrigues et al. (2022) showed that PDT using the photosensitizer aluminum-phthalocyanine mainly induced necrosis at the highest photosensitizer concentration and light dose, while milder protocols of PDT were able to efficiently induce ICD in both colorectal CT26 and mammary 4T1 murine adenocarcinoma cells [26]. In vitro, the milder PDT protocol consisted of 12.2 nM aluminum-phthalocyanine for colorectal adenocarcinoma CT26 cells and 9.0 nM aluminum-phthalocyanine for breast adenocarcinoma 4T1 cells, both irradiated with 25 J/cm^2^ red light dose. The exposure or release of calreticulin, HSP70, HSP90 and HMGB-1 close to that promoted by this PDT protocol was comparable to that observed with mitoxantrone treatment. Increasing the photosensitizer concentration and light dose resulted in a reduction on the release of these DAMPs. This shows that the PDT regimen has to be fine-tuned to induce ICD.

As commented previously, there are several ICD inducers described in the literature. Among them are classical approaches, such as chemotherapy, PDT and radiotherapy. These classical therapies are usually aimed at tumor destruction, mainly by inducing direct cytotoxicity to target tumor cells, regardless of the death mechanism underlying this effect. This cytotoxicity induced in a noncontrolled fashion can fail to trigger immune activation. Immune-targeted cytotoxic cancer treatments have to thus be designed to induce ICD, representing an adaptation in former protocols in order to increase the immunogenicity of cancer treatments.

Several ICD inducers are used in well-recognized therapies. Thus, we understand that the translation of this knowledge to clinical protocols can be easier if compared to the implementation of new anticancer drugs which are currently under development. As previously discussed, the literature shows evidence of abscopal effects in radiotherapy and PDT protocols, for example, that could be due to ICD. However, due to the complexity of cancer cells and tumor tissues, it is probable that personalized approaches are necessary to the successful use of immune-activating strategies in the clinic. Another barrier to the development of clinical protocols for ICD induction is the lack of standardization in ICD studies, as evidenced in this section. Thus, the use of widely recognized, bona-fide parameters that demonstrate that an immunogenic response is being induced is essential for translating research into clinical protocols [107]. This can be supported by guidelines proposed in the literature [29].

## 7. Delivery of DAMPs and ICD-Inducers to Tumor Tissues

As commented previously, ICD is triggered by the combined release of a specific set-up of DAMPs that includes calreticulin, HMGB-1, and ATP, among others. These molecules are exposed to the external cell membrane, and/or released to the external media in dying cells. All of these events are part of a cyclic process which is finely regulated during this specific sub-type of apoptosis. In terms of cell biology, as previously mentioned, several therapeutic procedures can trigger this event and have been investigated in both pre-clinical and clinical applications [88].

However, due to the variety of these different types of stimuli, the exposure or release of DAMPs can vary among the therapeutic strategies. Moreover, it is possible that, for clinical applications, the control of DAMPs exposure and release can be different among patients and tumor stages due to internal sub-tumoral tissue organizations and the different phenotypes of cancer cells. This situation makes the clinical translation difficult, delaying this therapy for patients.

To reduce this potential drawback, some authors have discussed alternative strategies to burst the DAMPs present inside tumor tissues (Figure 3), thus ensuring the presence of these molecular triggers for ICD induction. In terms of efficacy, the simple delivery strategy of DAMPs to target tumor tissues may not be efficient because ICD induction depends both on the release of chemotactic agents and on the presence of recognition molecules, such as calreticulin, on the cell membrane surface. Moreover, as discussed earlier, calreticulin exposure is a key factor for specific cell recognition.

Alternatively, some authors proposed the use of gene therapy to deliver DNA sequences that could increase DAMPs expression in target tumor cells. The rationale for this strategy is supported by the fact that some tumor types have a reduced expression of DAMPs, such as calreticulin. For instance, Garg et al. (2015) described a pre-clinical tumor model that reduces the constitutive calreticulin expression, thus reducing the effectiveness of immunogenic protocols [104]. The authors defined this tumor model as resistant to immunization against cancer cells. In terms of microevolution, it makes sense, as calreticulin is a molecule that activates tumor cell phagocytosis by APC. In this situation, selected tumor clones could reduce the expression of DAMPs, thus impairing ICD activation.

Due to the challenge of tumor targeting, nanoparticles or nanocarriers could be used to concentrate ICD inducers and DAMP molecules close to tumor tissues [26,27,108]. This is the classical argument for using nanotechnology for tumor therapy. Within this approach, nanocarriers are passively or actively delivered to the tumor regions and release the carried ICD activators to induce cell death and initiate the immune recognition and then the immune surveillance against malignant cells. This approach is somewhat different from the direct delivery of DAMPS to tumor tissues, but has been used successfully [108].

In the ICD approach widely proposed in the literature, conventional drugs, such as doxorubicin or mitoxantrone, for example, will lead tumor cells to succumb to ICD and release DAMPS. The preclinical results are promising; however, there are some concerns about the translation possibilities for this strategy, especially for the passive delivery of nanocarriers for tumor tissues. The main problems are the structural differences between preclinical induced tumor tissues and natural tumors that are developed in clinical conditions [109]. The argument is that passive targeting is not reproduced in clinical conditions. Despite all this discussion, there is some evidence that, at least in part, nanocarriers can increase the delivery of drugs and immunoadjuvant molecules to tumor tissues.

In the work by Zhou et al. (2022), murine melanoma (B16F10) cells were subjected to different treatments, such as hypoxia, cisplatin, radiotherapy, photodynamic therapy, and hypochlorous acid (HOCl). The cell-delivered secretions (CDS) of melanoma cells treated with HOCl activated dendritic cells and macrophages and produced the best antitumor immune response when compared to the other treatments. Aiming to increase the effectiveness of the treatment, the HOCl-CDS produced in vitro was then associated to nanofibers of a scaffold hydrogel containing melittin and RADA24 peptides. This nanosystem was then injected into the subcutaneous melanoma in vivo (C57BL/6 mice). The results indicated that the obtained hydrogel induced cell death, cytotoxicity in T lymphocytes, and increased the antitumor effect of the immune checkpoint inhibitors.

Another example of a successful use of nanocarriers for increasing the immunogenicity of tumors was published by Sethuraman et al. (2020) [105]. In this work, the authors describe a liposome nanocarrier with a DNA plasmid encoding calreticulin. They observed that this strategy increased the expression of calreticulin in target tumor cells, reducing tumor growth due to immune activation. Interestingly, when they combined this liposomal formulation with the application of focused ultrasound treatment, the results were better. This improvement is probably related to the temperature increase provided by the ultrasound. In higher temperatures, some amount of cell death could be triggered, which in combination with calreticulin superexpression could increase the immune activation.

The authors did not evaluate the modality of cell death induced by the treatment. However, it shows the potential of DAMPs delivery for immune system activation. As a potent phagocytosis inducer, calreticulin is a key factor for alternative types of immune activation. As noted previously, this protein was included as a molecular signature for the ICD, but its presence in external spaces may also promote tumor recognition by APC, thus contributing to immunological surveillance [106].

In this section, we presented potential uses of drug delivery strategies to optimize ICD induction. First, nanoparticles could deliver DAMPs directly to target tissues; second, nanocarriers could deliver ICD inducers; and third, nanoparticles could be used during gene therapy to increase the expression of DAMPS in the target tumor tissues. There is certainly no ideal strategy, but an effective therapy could be based on the combination of different approaches, and eventually on other strategies to optimize ICD induction in immunotherapies.

## 8. Conclusions

Although the induction of ICD has not been rationally used as a clinical treatment modality, it has potential for being exploited as an immunotherapy tool. The recent advances in immunotherapy have opened new fronts and possibilities in the fight against melanoma, for instance, with prolonged survival and durable responses in many patients. Studies on the immune evasion strategies deployed by tumors, as well as on the mechanisms underlying the activation of immune cells against abnormal host cells, can help to develop new weapons against cancer. The current knowledge allows us to suggest that the induction of ICD combined with other immunotherapeutic strategies, such as immune checkpoint blockade, can be explored to treat cancer. However, barriers such as the lack of standardization of ICD detection in clinical patients, as well as the need for more personalized protocols based on the detailed characterization of the tumor cells, have to be overcome to translate experimental protocols into the clinic.

## Figures and Tables

**Figure 1 pharmaceutics-14-01564-f001:**
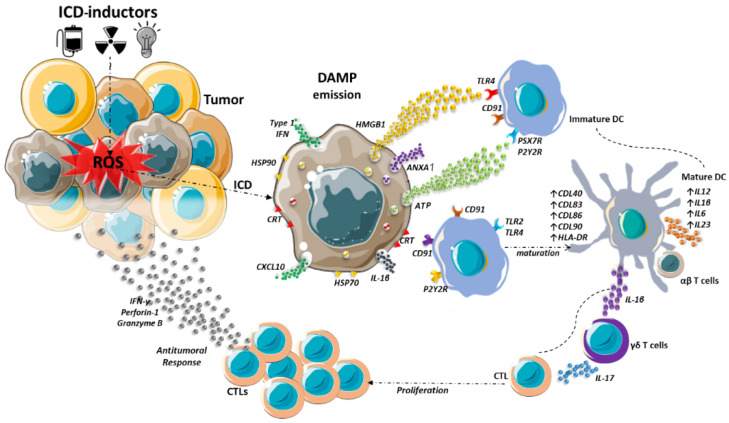
Engagement of adaptive immune response after immunogenic cell death in a tumor. Chemotherapy, radiotherapy, and photodynamic therapy can induce immunogenic cell death (ICD), which is a programmed cell death accompanied by the exposure of damage-associated molecular patterns (DAMPs). This can occur, for instance, as a consequence of oxidative stress in the endoplasmic reticulum. Some DAMPs, such as heat-shock protein (HSP)70, HSP90, and calreticulin are exposed on the plasma membrane, while others such as adenosine triphosphate (ATP), high mobility group box 1 protein (HMGB1), C-X-C motif chemokine ligand 10 (CXCL10) and annexin A1 (ANXA-1) are released to the extracellular medium. DAMPs then activate pattern recognition receptors of dendritic cells (DCs) and other antigen-presenting cells. This culminates in the maturation of the DCs and in the recruitment and activation of T cells. In this way, ICD can trigger or boost an adaptive antitumor immune response.

**Figure 2 pharmaceutics-14-01564-f002:**
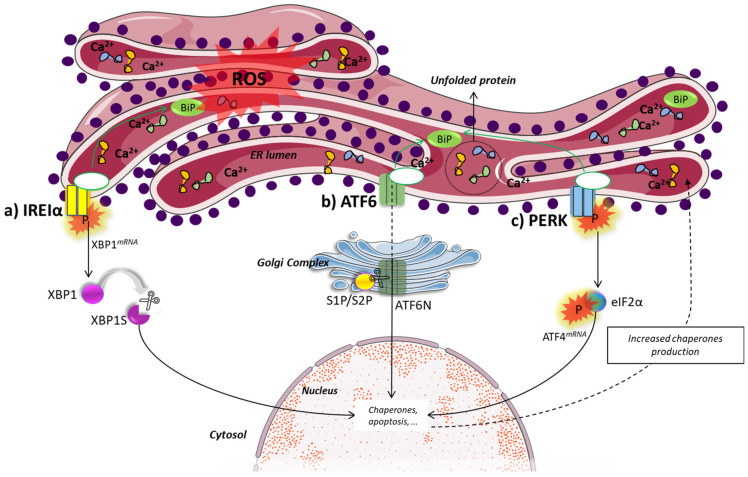
Activation of responses to protein malformation. These three pathways—IRE1α (**a**), ATF6 (**b**) and PERK (**c**)—are activated in unfolded protein response (UPR) to cope with disturbances in protein folding and to restore endoplasmic reticulum homeostasis after stress. The phosphorylation of eIF2α is a hallmark of immunogenic cell death. Legend: ROS: reactive oxygen species; BiP: Binding immunoglobulin protein; ER: endoplasmic reticulum; IRE1α: inositol-requiring enzyme 1α; XBP1: X box-binding protein 1 mRNA; XBP1S: spliced X box-binding protein 1 mRNA; ATF6: activating transcription factor 6; PERK: protein kinase RNA-like endoplasmic reticulum kinase; ATF4: activating transcription factor 4; S1P: site-1 protease; S2P: site-1 protease; eIF2α: eukaryotic initiation factor 2α.

**Figure 3 pharmaceutics-14-01564-f003:**
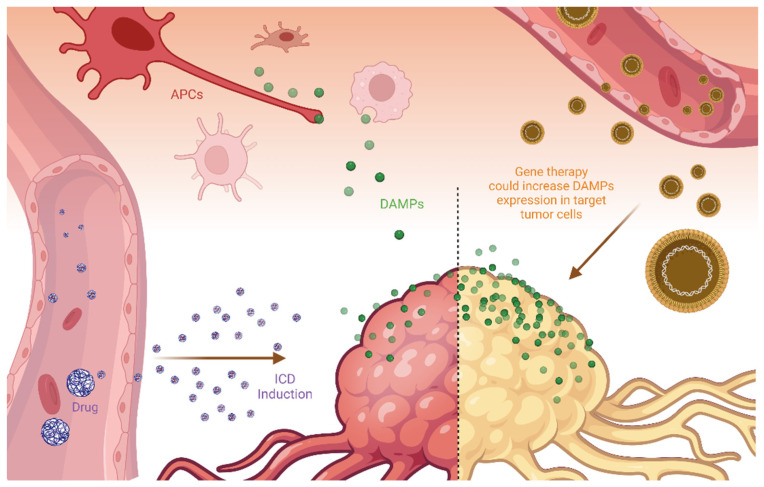
The induction of immunogenic cell death (ICD), the exposure of damage-associated molecular patterns (DAMPs), and the activation of antigen-presenting cells (APC) can be enhanced by delivery strategies based on nanotechnology.

**Table 1 pharmaceutics-14-01564-t001:** Damage-associated molecular patterns involved in immunogenic cell death.

DAMP	Abbreviation	Effect on Immune Cells	References
Annexin A1	ANXA1	Expressed in different cells (neutrophils, eosinophils, and monocytes), ANXA1 has a role in the regulation and resolution of inflammation. It can act as a negative regulator of innate immunity, with neutrophils being its main target; it activates the migration of APCs towards the dying cells, facilitating their engulfment and processing.	[34,74]
Adenosine triphosphate	ATP	ATP acts as a strong chemoattractant and promotes not only the recruitment of immune cells but also their maturation.	[73,75]
Calreticulin	CRT	Calreticulin acts as phagocytosis inducer. Its exposure and the release of ANXA1, ATP and HMGB1 result in the attraction and maturation of DCs in the tumor microenvironment.	[36,39]
Deoxyribonucleic acid	DNA	With its accumulation in the cytoplasm, DNA can stimulate innate immune responses.	[93]
High mobility group box 1 protein	HMGB1	Acts as an essential DAMP in the DCs activation, stimulating the production of pro-inflammatory factors, strongly contributing to the immunogenicity of ICD.	[29,94]
Heat-shock protein	HSP 70 HSP 90	HSP act as eat-me signals for phagocytes. They can induce DC maturation and promote target engulfment by APC cells.	[29]
Type I interferon	IFN-I	IFN-Is acts as potent immunostimulatory proteins and have a crucial role in ICD. It can modulate the maturation, differentiation, and migration of DC cells, increase primary antibody responses, and activate B and T cells directly or indirectly.	[29,71]
Ribonucleic acid	RNA	It recruits leukocyte and M1-type macrophages.	[95]
Uric acid	UA	Crystalline UA can produce inflammatory mediators through macrophage activation and the enhancement of T cells.	[96]

## Data Availability

Not applicable.

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
