# Peer review of "An Overview on Immunogenic Cell Death in Cancer Biology and Therapy"

_pharmaceutics, 2022, doi:10.3390/pharmaceutics14081564_

Round 1

Reviewer 1 Report

This manuscript overviews different aspects of immunogenic cell death (ICD) in cancer biology and treatment, and discusses how ICD can help to trigger or boost immune responses against cancer. This review is well prepared, and has well described the ICD in cancer biology and therapy. However, the author needs to address the following issues.

1. In this section of Cancer immune surveillance and immunotherapy, the authors need to streamline their development of immunosurveillance and focus on immune checkpoint inhibitors mentioned in the first paragraph.

2. The relationship between different types of cell death and ICD should be briefly discussed, such as Ferroptosis and Pyroptosis.

3. Which types of cell treatment will lead to more generation of ICD? The author can give a short discussion (e.g. Zhou et al. Bioactive materials 9 (2022): 541-553).

4. Barriers to clinical translation of ICD for cancer immunotherapy should be discussed.

5. The authors need to discuss the reasons why ICD is not used as a cancer treatment modality in the conclusion section.

6. In Figure 1, the format of ANXA-1 and ATP font needs to be adjusted .

7. In Figure 2, authors need to annotate the abbreviations for readers to understand.

8. The authors need to revise the last paragraph of the Endoplasmic reticulum stress section because it is abrupt.

9. The authors need to add a table about the types of DAMPs to be more intuitive.

10. Some relevant references about ICD should be cited, such as Sun et al. Asian journal of pharmaceutical sciences 16.2(2021): 129-132; Fucikova et al. Cell death & disease 11.11 (2020): 1-13.

Author Response

Dear reviewer,

In behalf of all the authors, I want to thank you for your precious comments and suggestions.

The manuscript is now far better than before your contribution.

Best regards,

Luís A. Muehlmann

Answers to reviewer #1

Comments and Suggestions for Authors

This manuscript overviews different aspects of immunogenic cell death (ICD) in cancer biology and treatment, and discusses how ICD can help to trigger or boost immune responses against cancer. This review is well prepared, and has well described the ICD in cancer biology and therapy. However, the author needs to address the following issues.

  1. In this section of Cancer immune surveillance and immunotherapy, the authors need to streamline their development of immunosurveillance and focus on immune checkpoint inhibitors mentioned in the first paragraph.

A.: Thanks for the suggestion, but in this section, we intended to present a historical overview on the developments that led to cancer immunotherapy, rather than discussing cancer immunosurveillance and immune checkpoint inhibitors. We think that the former title of this section wasn’t fit to this purpose. Thus, we changed it in the revised text.

  1. The relationship between different types of cell death and ICD should be briefly discussed, such as Ferroptosis and Pyroptosis.

A.: The second paragraph in the section “Immunogenic cell death” now is dedicated to this discussion.

  1. Which types of cell treatment will lead to more generation of ICD? The author can give a short discussion (e.g. Zhou et al. Bioactive materials 9 (2022): 541-553).

A.: We thank you for this suggestion. We have already discussed along the text, mainly on the Sections “Immunogenic cell death” and “ICD and DAMPs in cancer therapy”, the conditions that are required to induce ICD. 

  1. Barriers to clinical translation of ICD for cancer immunotherapy should be discussed.

A.: Thank you for the suggestion. Indeed, this is an important matter, and we had already discussed it in some paragraphs in the section “ICD and DAMPs in cancer therapy”. Moreover, we had now added further discussion on this issue to this same section (highlighted in yellow).

  1. The authors need to discuss the reasons why ICD is not used as a cancer treatment modality in the conclusion section.

A.: New text, highlighted in yellow, was added to the Conclusion section to meet this demand.

  1. In Figure 1, the format of ANXA-1 and ATP font needs to be adjusted.

A.: The names were corrected.

  1. In Figure 2, authors need to annotate the abbreviations for readers to understand.

A.: Thank you for noticing this mistake. It was corrected.

  1. The authors need to revise the last paragraph of the Endoplasmic reticulum stress section because it is abrupt.

A.: This section now ends introducing the next one.

  1. The authors need to add a table about the types of DAMPs to be more intuitive.

A.: The table was added to the manuscript.

  1. Some relevant references about ICD should be cited, such as Sun et al. Asian journal of pharmaceutical sciences 16.2(2021): 129-132; Fucikova et al. Cell death & disease 11.11 (2020): 1-13.

A.: Thank you for this quite pertinent suggestion. These references are now cited in our manuscript.

Reviewer 2 Report

Rodrigues and Morais et al. provided an overview of immunogenic cell death (ICD) in cancer biology and therapy. This review is organized and well written and should attract the readership of Pharmaceutics working in cancer immunotherapy and related fields. I would recommend its acceptance for publication after addressing these issues.

1.     It would be great if the authors could provide one or two figures in the “ICD and DAMPs in cancer therapy” and “DAMP delivery to tumor tissues” sections to better illustrate the content of these sections.

2.     The authors may want to dedicate a section/subsection to discuss the use of nanomaterials/nanocarriers, either independently or in conjunction with other approaches, to induce ICD.

3.     The authors should provide more specific directions that can be explored in the future to accelerate the use of ICD in the clinic.

Author Response

Dear reviewer,

In behalf of all the authors, I want to thank you for your precious comments and suggestions.

The manuscript is now far better than before your contribution.

Best regards,

Luís A. Muehlmann

Answers to reviewer #2

Comments and Suggestions for Authors

Rodrigues and Morais et al. provided an overview of immunogenic cell death (ICD) in cancer biology and therapy. This review is organized and well written and should attract the readership of Pharmaceutics working in cancer immunotherapy and related fields. I would recommend its acceptance for publication after addressing these issues.

  1. It would be great if the authors could provide one or two figures in the “ICD and DAMPs in cancer therapy” and “DAMP delivery to tumor tissues” sections to better illustrate the content of these sections.

A.: A new figure was added to the “DAMP delivery to tumor tissues”

  1. The authors may want to dedicate a section/subsection to discuss the use of nanomaterials/nanocarriers, either independently or in conjunction with other approaches, to induce ICD.

A.: The use of nanotechnology for inducing ICD was included in the reconfigured section “Delivery of DAMP and ICD-inducers to tumor tissues”, as suggested. The new text is highlighted in yellow.

  1. The authors should provide more specific directions that can be explored in the future to accelerate the use of ICD in the clinic.

A.: Thank you for the suggestion. Indeed, this is an important matter, and we had already discussed it in some paragraphs in the section “ICD and DAMPs in cancer therapy”. Moreover, we had now added further discussion on this issue to this same section (highlighted in yellow).